# Validation of CHERG'S Verbal Autopsy-Social Autopsy (VASA) tool for ascertaining determinants and causes of under-five child mortality in Pakistan

**Muhammad Bilal Siddiqui**[1], **Chiu Wan Ng**[1]*, **Wah Yun Low**[2], **Khadijah Abid**[3]

1 Department of Social and Preventive Medicine, Faculty of Medicine, University of Malaya, Kuala Lumpur, Malaysia, 2 Dean's Office, Faculty of Medicine, University of Malaya, Kuala Lumpur, Malaysia, 3 Department of Public Health, Faculty of Life Sciences, Research Villa, Shaheed Zulfiqar Ali Bhutto Institute of Science and Technology (SZABIST), Karachi, Pakistan

* chiuwan.ng@ummc.edu.my

**Data Availability Statement:** The datasets used and analysed during the current study are available on the following link. https://figshare.com/s/883fe45ea7da71483acd.

## Abstract

The majority (40%) of the world's under-five mortality burden is concentrated in nations like Nigeria (16.5%), India (16%), Pakistan (8%), and the Democratic Republic of the Congo (6%), where an undetermined number of under-five deaths go unrecorded. In low-resource settings throughout the world, the Verbal Autopsy-Social Autopsy (VASA) technique may assist assess under-five mortality estimates, assigning medical and social causes of death, and identifying relevant determinants. Uncertainty regarding missing data in high-burden nations like Pakistan necessitates a valid and reliable VASA instrument. This is the first study to validate Child Health Epidemiology Reference Group-CHERG's VASA tool globally. In Pakistan, data from such a valid and reliable tool is vital for policy. This paper reports on the VASA tool in Karachi, Pakistan. Validity and reliability of the CHERG VASA tool were tested using face, content, discriminant validation, and reliability tests on one hundred randomly selected mothers who had recently experienced an under-five child death event. Data were computed on SPSS (version-21) and R software. Testing revealed high Item-content Validity Index (I-CVI) (>81.43%); high Cronbach's Alpha (0.843); the accuracy of between 75–100% of the discriminants classifying births to live and stillbirths; and I-CVI (>82.07% and 88.98% respectively) with high accuracy (92% and 97% respectively) for assigning biological and social causes of child deaths, respectively. The CHERG VASA questionnaire was found relevant to the conceptual framework and valid in Pakistan. This valid tool can assign accurate medical and non-medical causes of child mortality cases occurring in Pakistan.

## Introduction

Verbal autopsy (VA) is a technique that involves structured questions focusing on the signs and symptoms the deceased case faced during the illness but before the mortality incident. VA is used to record various relevant biological determinants (BDs) linked with child mortality

**Funding:** The author(s) received no specific funding for this work.

**Competing interests:** The authors have declared that no competing interests exist.

**Abbreviations: C.A.**, Cronbach's Alpha; **C.C.**, Canonical correlation; **C.V.**, Content validation; **CHERG**, Child Health Epidemiology Reference Group; **CoD**, Cause of Death; **CRVS**, Civil Registration and Vital Statistics; **D.V.**, Discriminant Validation; **E.V.**, Eigenvalue; **HMIS**, Health Management Information System; **I-CVI**, Item-content Validity Index; **LR**, Likelihood-Ratio (LR) test; **MCoD**, Medical Cause of Death; **PCVA**, Physicians certified VA technique; **R.A.**, Reliability Analysis; **SA**, Social Autopsy; **SCoD**, Medical Cause of Death; **SPSS**, Statistical Package for Social Sciences; **TPtoSCF**, The Pathway to Survival Conceptual Framework; **VA**, Verbal Autopsy; **VASA**, Verbal autopsy/Social Autopsy; **W.L.**, Wilk's Lambda; **WHO**, World Health Organization.

incidents, and based on such information, VA helps in assigning the medical cause of death (MCoD). Similarly, social autopsy (SA), a relatively new technique with a questionnaire structure similar to VA, helps record non-biological determinants (especially the health-seeking process, and issues and barriers in the process of accessing health care during the child illness process but before the mortality event) so that the social cause of death (SCoD) of any child death can be concluded. VA has been used in low-resource settings when the cause of any child death (CoD) has not been ascribed by a health care provider [1–3]. In contrast to VA, SA is not typically used for child mortality in the majority of developing countries, especially Pakistan. Both VA and SA not only provide the MCoD and SCoD to the child mortalities respectively but also assist in updating epidemiological estimates on the mortality which have been missed to be recorded by the health system in these geographies [4, 5]. Although VA has been used in nations with little resources, there is still a need to investigate a wider range of data on the societal factors that contribute to child mortality [3, 6]. Hence, in a more recent timeframe, a concept of running through the missed mortality case with a VA and SA tool has been attempted in some countries (Cameroon [6, 7], Mozambique [8], Niger [9–11], Uganda [12–15], Rwanda [16, 17], Malawi [6], Nigeria [18], Guinea [19]) which helped to get detailed evidence of social causes of different medically assigned CoDs. However, these attempts made use of different VA and SA tools in combinations [20].

The Child Health Epidemiology Reference Group (CHERG) was established in 2001 with the support of WHO and UNICEF to provide technical guidance to relevant institutions and organisations for collecting evidence on determinants and causes of child mortality and improving global child survival epidemiological estimates [7, 21]. The agency developed an integrated VASA tool based on a most holistic approach for recording different biological determinants (child-specific, maternal-specific, determinants across the CoC, and preventive measures taken before and during the pregnancy and child illness) and non-biological determinants (NBDs) identified through several models (The Three Delay Model), conceptual framework (TPtoSCF); and several recommendations for VA suggested by the World Health Organization [22] to record different biological and non-biological determinants affecting the under-five mortality [20, 23–25]. In addition to recording biological determinants, this tool also focuses on all the barriers which come across as hindrances throughout the health-seeking process when the child was ill [21]. This tool is one of the assets that can be utilized to acquire information on the relevant biological and non-biological determinants of child mortality for every missed child mortality case on a single visit to the household and has already supported several countries to gather evidence on the medical and social causes to child deaths [10, 20, 26]. The extensive data obtained from CHERG VASA tool based investigations provides highly elaborated evidence and information on extended determinants of child mortality which is of great public health importance [4, 10, 26].

Similar to any data collection tool, it's always been a concern whether the data collected through VA and SA are correct, reliable, and represent the information that the researcher intends to collect. The process of validation involves testing the questionnaire on validity and reliability parameters. To obtain the best possible data, many efforts have been undertaken globally in refining the methodology of VA [27–33]. In continuation of such efforts, a few validation studies have been undertaken globally as well in Pakistan [34] to assess the capacity of VA tools in providing precise and reliable evidence on the causes of child mortality [27, 28, 32, 33]. However, no attempts have been made to validate CHERG's integrated VASA tool. Considering the current status of child mortality estimates in Pakistan [34–36], CHERG's integrated VASA tool was validated to gather evidence on a comprehensive list of medical and social determinants and medical and social causes of under-five mortality in Pakistan, where child mortality cases may be missed to be registered under national CRVS and Health

Management Information System. This is the first time a VASA integrated tool has been validated so thoroughly, concentrating on most of the approaches used to evaluate validity and reliability. We expect the validated tool would provide comprehensive data on under-five mortality in Pakistan. This paper presents the results of the validation of the CHERG VASA tool. The validated instrument will be used to conduct a future VASA child mortality survey in Pakistan's underprivileged community.

## Materials and methods

This validation study is a component of the *Karachi VASA child mortality study* [37] and is designed to objectively assess the validity and reliability estimates of the CHERG's VASA questionnaire, which was later utilized in *Karachi VASA child mortality study*.

### The sample size

Although there are no specific guidelines for the sample size of respondents required for pre-survey validation of the VASA tool [38], the only available guideline suggests [39] that, in general, the sample size required for any questionnaire validation should be a minimum of 20% of the total sample size of the study. Since 400 under-5 died cases were included in the Karachi VASA child mortality study, 20% of 400 cases (80 cases) were selected from the study sites to participate in the questionnaire validation study. The original sample size of 80 was increased to 100 to overcome the issue of potential data entry errors. Mothers of these 100 under-five deceased cases (n = 100) were selected from 12 selected sites i.e., Cheneser goth (CG), Mujahid colony (MC), Haryana Colony (HC), Sultanabad (SD), Bilal colony (BC), Sherabad (SA), Yaseenabad (YS), Landhi slums (LS), Khuda ki Basti (KKB), Safora goth (SG), Malir slum (MS), and Sultanabad (SD), based on the number of deaths occurred in these sites during the past one year (from June 1, 2015, to May 3, 2016) before the data collection. These 12 study sites were located in five of the city's districts, into which the city was subdivided at the time of the study, namely, District Malir, District Central, District South, District West, District East, and District Korangi. Two study sites (one urban and another one urban-slum) were randomly selected from each of the six districts, making a total of 12 study sites for research purposes. The urban sites and the urban-slum sites were classified based on the definition of a slum from UN-Habitat [40]. To ensure that the sample was representative of the entire population, the sample (100 mothers of deceased under-five deaths, falling under strict inclusion and exclusion criteria) was drawn from all six districts of Karachi.

### Sampling methodology

The estimates of all the under-five deaths (occurred between 01st June 2015 till 03rd May 2016) from each of the study sites were identified from each site through multiple sources (household visits i.e., snowball sampling; local graveyards; local town health and union council offices). To fulfil the sample size of 100 respondents for the validation study, each identified study site was allocated a quota based on the **proportionate quota sampling technique** (proportionally of the overall sample size required for the study). The details (quota calculation) are available in Table 1.

### Inclusion and exclusion criteria

To reduce response bias, only women whose children under the age of five passed away during the previous year (from 01st June 2015 till 3rd May 2016) were included in this validation study.

**Table 1. Sampling quota calculation.**

| S. No | STUDY SITE | SITE CODE | TOTAL IDENTIFIED DEATHS IN EACH SITE | (DEATH IN EACH SITE/TOTAL DEATHS IN 12 SLUMS) | QUOTA TO BE ALLOCATED FOR EACH SITE FOR VALIDATION |
|---|---|---|---|---|---|
| | | | (A) | (B) | (C) |
| 1 | Cheneser goth | CG | 210 | 210/1926 | 11 |
| 2 | Mujahid colony | MC | 168 | 168/1926 | 9 |
| 3 | Hijrat colony | HC | 159 | 159/1926 | 8 |
| 4 | Haryana colony | HC | 57 | 57/1926 | 3 |
| 5 | Sultanabad | ST | 111 | 111/1926 | 6 |
| 6 | Bilal colony | BC | 189 | 189/1926 | 10 |
| 7 | Sherabad | SA | 243 | 243/1926 | 13 |
| 8 | Yaseenabad | YS | 183 | 183/1926 | 9 |
| 9 | Landhi slum | LS | 150 | 150/1926 | 8 |
| 10 | Khuda ki Basti | KKB | 126 | 126/1926 | 6 |
| 11 | Safora goth | SG | 183 | 183/1926 | 9 |
| 12 | Malir slum | MS | 147 | 147/1926 | 8 |
| | Total | | 1926 | | 100 |

After being informed of the study's purpose and procedures, all participants were asked if they would be willing to participate and informed written consent was obtained. Those who did not wish to participate in the study were given the option to withdraw themselves from the study, and this was done with respect. All responses were treated as strictly confidential. It was ensured that, if a respondent experienced any emotional disturbance during the data collection process, the respondent would be referred to a health professional; however, no such case was encountered.

## Structure of the CHERG's VASA questionnaire

The CHERG's VASA questionnaire [23] is a semi-quantitative tool for collecting data on the signs and symptoms of illness (including varied BDs) and the relevant history of social circumstances (NBDs) right from the conception of the mother until the death of the ill child including few open-ended questions i.e., the open narratives (which specifically asks for any additional notes to be mentioned by the respondents in relation to the Verbal autopsy component as well as the summary from the interviewer of the whole case). It also quantitatively captures the signs and symptoms a child encountered during the illness and the social history of the deceased child before the death of the child. The initial pre-validated CHERG VASA questionnaire consisted of VA and SA components having 14 different modules (sections) focusing on the different variables ranging from the demographic profile of the respondent, family of the deceased child, parents of the deceased child, healthcare-seeking process and socioeconomic status to variables encompassing the entire continuum of care during the pregnancy till the death of the child.

After the section on the VASA general information (which collects information on the general variables), the VA component has 06 modules altogether, while the SA component has 07 modules (Table 2, columns 1&2). These VA, SA, and pertinent modules have been organized in such a way that as the respondent's interview progresses, the chronology of events becomes

**Table 2. Structure of CHERG's VASA questionnaire.**

| Details of different modules of VA & SA components included in the VASA tool | | | Sequence of different modules of VA & SA components in VASA tool |
|---|---|---|---|
| **VASA General Information:**<br>• Deceased background,<br>• Interview details,<br>• Consent,<br>• Respondent's details,<br>• Details of others present during the interview | **VA Sections:**<br>VA-1: Background<br>VA-2: Maternal History<br>VA-3: Neonatal deaths<br>VA-4: Infants and child deaths<br>VA-5: Health records<br>VA-6 Open-ended questions (any additional information about illness and death) | **SA Sections:**<br>SA-1: Mother and her household<br>SA-2: Social capital<br>SA-3: Maternal history<br>SA-4: Care-seeking for maternal complications<br>SA-5a: Care of the newborn<br>SA-5b: Preventive care of post-neonates<br>SA-6: Care-seeking of child's fatal illness<br>SA-7: Open-ended questions (any additional information about social circumstances) | (VA-1); (SA-3); (VA-2); (SA-4); (SA-5a); (VA-3); (SA-5b); (VA-4); (SA-6); (VA-5); (SA-1); (SA-2); (VA-6) & (SA-7). |

clear to the researcher (Table 2, column 3). To record relevant information on relevant age groups (i.e., neonatal deaths, and post-neonatal deaths), specific skip patterns exist in the tool.

## Translation & back translation

The CHERG VASA tool is freely available [23]. The tool is available either in the integrated version of VASA or in different sections separately. Before validation, the questionnaire was forward translated into the local language, Urdu, and then back-translated to English based on the **"Brislin Back-translation"** technique [41]. The process involved a professional language translator and a panel of professional bilingual subject experts (paediatricians, neonatologists, gynaecologists, obstetricians, social scientists, and psychologists). The final translated version was processed for the validation process.

## Data collection for the validation study

The respondents (mothers of deceased children) were recruited, and their written consents were obtained. A total of 100 VASA interviews were conducted. Data was recorded on both the platforms i.e., Paper-And-Pencil Interviews (PAPI)(i.e., paper-based VASA tool); and Computer Assisted Personal Interview software (CAPI)(an electronic platform format for data collection with skip patterns using CSPro software application) and were compared to check any inconsistencies. However, no inconsistencies were encountered between the PAPI and CAPI platforms. The completion rate of the questions on both PAPI and CAPI was the same (100%), however, there were some issues by the interviewee and interviewer on the implementation of these two methods which are elaborated in a qualitative study [42] The only difference between the two platforms was the ease of data collection and less time utilized for data collection by the data collectors [42]. Data quality was checked by a quality management team and researcher. None of the mothers refused or rejected the interview. Extensive training to the data collectors and data entry operators (with a minimum educational background of having Intermediate level (Higher Secondary School Certificate i.e., Grade-12) was given on communication skills, details of the project, questionnaire design and data entry process. Collected data were shared with a panel of subject specialists (five Paediatricians with expertise in Neonatology) trained for PCVA for reviewing the data and assigning the MCoD and SCoD (five sociologists) through VA and SA components of the integrated VASA tool. Medical CoD was assigned by the physicians (five Paediatricians with expertise in Neonatology) (through Physician Certified Verbal Autopsy-PCVA), where the PCVA trained physicians reviewed the mortality case reports (complete VASA data) including signs and symptoms, open narratives,

medical history and sequence of events prior to death of deceased children and assigned the most probable MCoD. Initially the data was shared with one reviewer, who assigned a MCoD. Once received, the data was then shared with second reviewer and the level of agreement was checked. In case of any disagreement between the initial two reviewers, a third reviewer was assigned until the agreement between the two different reviewers was achieved. The MCoD assigned by the reviewers were kept confidential by the other reviewers. While conducting PCVA, reviewers were kept blinded with the MCoD assigned on the death certificates of those cases (under-five mortalities) with a record of death certificates. Similarly, the SCoD were assigned by the social scientists through the same process after reviewing the integrated VASA tool.

## Tool's validation procedure

As illustrated in Table 3 and Fig 1, the CHERG VASA tool was evaluated on its *validity* and *reliability* through face validation, content validation, discriminant validation, and reliability analysis and the details are described in the respective sections below.

## Results

The sample of 100 respondents and the included households had the following demographic characteristics (Table 4). Only key demographics appear in Table 4. Full details are available can be accessed upon request.

## Characteristics of included deceased children

The Table 5, mentions the details of some of the important characteristics of the deceased children included in the validation study.

**(a) Face validity assessment.** For assessing the face validity of the tool, the questionnaire was reviewed and critically evaluated by a panel of 6 experts—a researcher and 5 expert(s) in the relevant field of interest (e.g., paediatrician, gynaecologist, obstetrician, social scientist, and psychologist [44, 45] on the tool's format to find out, whether all the sections and items be easily readable, clear, and straightforward to understand and comprehend by the interviewees and respondents; and on the need of rephrasing, omitting, or adding any questions. The panellists found no need for any modifications in terms of rephrasing questions, omitting any option of any item, adding an option of any item, etc. The team identified that the items of the

**Table 3. Summary of methodology for validation of CHERG's VASA questionnaire.**

| Assessment | Method | Outcome |
|---|---|---|
| **Face Validation** | **Subjective review** of the tool by a panel of experts | The updated questionnaire which was assessed on C.V., D.V., & R.A. (below in this table) |
| **Content Validation (C. V.)** | Calculation of **I-CVI** using agreement level of 05 experts on a dichotomous scale | I-CVI ranged between 81.43% to 96.36%. |
| **Discriminant Validation (D.V.)** | Calculation using **Eigenvalue (E.V.)**, **Canonical correlation (C.C.)**, **Wilk's Lambda (W.L.)**, and **Accuracy** | • E.V. = between 0.61 to 735.11<br>• C.C. = between 0.86 and 1.00<br>• W.L. = <0.001<br>• Accuracy of >75.07% |
| **Reliability Analysis (R. A.)** | Calculation using **Cronbach's Alpha (C.A.)** statistics | C.A ranged between 0.843 and 0.973 |

**C.V.**: Content validation; **D.V.**: Discriminant Validation; **R.A.**: Reliability Analysis; **I-CVI**: Item-content Validity Index; **E.V.**: Eigenvalue; **C.C.**: Canonical correlation; **W.L.**: Wilk's Lambda; **C.A.**: Cronbach's Alpha;

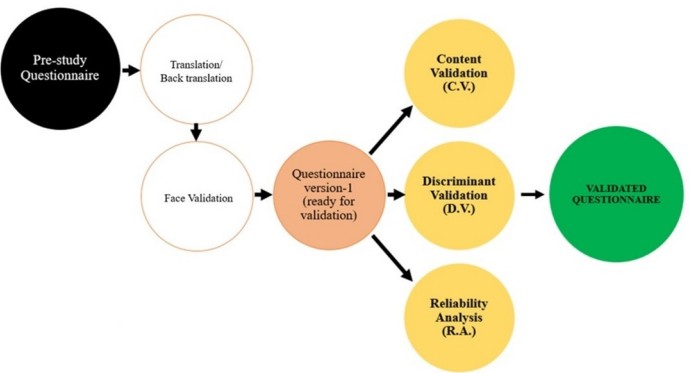

**Fig 1. Flow chart of the validation process.**

Table 4. Demographic characteristics of the enrolled households for validation study.

| Characteristics | n | % |
|---|---|---|
| **Mothers' characteristics** | | |
| | 100 | 100 |
| Married or living with a man | 97 | 97 |
| Widowed | 3 | 3 |
| **Mother's mean age when first married (years):** | | |
| 15–20 | 8 | 8 |
| 21–24 | 54 | 54 |
| 25–28 | 23 | 23 |
| 29–32 | 11 | 11 |
| 33–36 | 4 | 4 |
| **Mother years of schooling (mean years of schooling):** | | |
| Didn't attend school | 12 | 12 |
| Primary (1–5 years) | 40 | 40 |
| Middle (6–8 years) | 17 | 17 |
| Secondary (9–10 years) | 26 | 26 |
| Intermediate (11–12 years) | 5 | 5 |
| >12 years | 0 | 0% |
| **Main breadwinner** | | |
| Father | 46 | 46 |
| Mother | 31 | 31 |
| Both | 23 | 23 |
| **Household size** | | |
| <5 | 24 | 24. |
| ≥5 <10 | 66 | 66 |
| ≥10<15 | 10 | 10 |

*Improved sanitation facilities are likely to ensure hygienic separation of human excreta from human contact. They include flush/pour flush (to the piped sewer system, septic tank, pit latrine), ventilated improved pit (VIP) latrine, pit latrine with slab, and composting toilet [43]

**Table 5. Characteristics of the enrolled deceased under-five children for the validation study.**

| | N | % |
|---|---|---|
| **Gender of the deceased children** | | |
| Boy | 59 | 59 |
| Girl | 41 | 41 |
| **Age of deceased children** | | |
| Stillbirths | 33 | 33 |
| Livebirths | 67 | 67 |
| **Place of birth of the deceased children** | | |
| Government hospital | 17 | 17 |
| Private hospital | 41 | 41 |
| Home | 42 | 42 |
| **Age at which child died** | | |
| Born dead | 67 | 67 |
| Neonate | 17 | 17 |
| 1 month-59 month | 16 | 16 |
| **Frequency of different medical causes of under-five deaths** | | |
| Preterm delivery (P07) | 7 | 7 |
| Birth asphyxia (P21) | 4 | 4 |
| Neonatal Pneumonia (P23) | 2 | 2 |
| Neonatal Tetanus (A33) | 2 | 2 |
| Neonatal Sepsis (P63) | 1 | 1 |
| Neonatal Meningitis (G03) | 1 | 1 |
| Neonatal Congenital malformations (Q89) | 0 | 0 |
| Neonatal Sudden unexplained death (R99) | 0 | 0 |
| Neonatal Diarrhoea (A09) | 0 | 0 |
| Postneonatal Diarrhoea (A09) | 5 | 5 |
| Postneonatal Pneumonia (J22/J18) | 4 | 4 |
| Postneonatal Severe Malnutrition (D64) | 3 | 3 |
| Postneonatal Hemorrhagic fever (A99) | 2 | 2 |
| Postneonatal Meningitis (E46) | 1 | 1 |
| Postneonatal Malaria (B54) | 1 | 1 |
| **Mother's hospitalization** | | |
| Yes | | |
| No | | |
| **Place of birth** | | |
| Government hospitals | 46 | 46 |
| Private hospitals | 37 | 37 |
| Home | 17 | 17 |
| **Antenatal history** | | |
| **Attended at least antenatal visit in HCF** | | |
| 1 visit | 6 | 6 |
| 2 visits | 24 | 24 |
| 3 visits | 14 | 14 |
| 4 visits | 16 | 16 |
| No visit | 40 | 40 |
| **Child's hospitalization** | | |
| Yes | 11 | 11 |
| No | 89 | 89 |

*(Continued)*

**Table 5.** (Continued)

|  | N | % |
|---|---|---|
| **Child's hospital records** | | |
| **Yes** | **31** | **31** |
| **No** | **69** | **69** |
| **Which health care provider was consulted by mother during ANC** | | |
| **Did not see any HCP for ANC** | **41** | **41** |
| **Local dai** | **46** | **46** |
| **Private clinic** | **13** | **13** |

HCP: Healthcare professional, ANC: Antenatal care

tool were phrased in such a way that the respondent would need to think before responding, thereby minimizing the chances of response bias. The translated questionnaire which went for face validation was then named **"VASA Questionnaire-version-1".** This version of the questionnaire was utilized for content validation as well as to collect data for undertaking discriminant validation.

**(b) Content validity assessment.** It involved a panel of five subject specialists (paediatrician, gynaecologists, obstetrician, social scientist, and psychologist) who gave their ratings on different parameters (especially on the comprehensiveness of the tool and the relativity of the items on the tool with the background conceptual framework of the topic). Each member from a panel was requested to rate every item on the VASA Questionnaire-version-1 on a 5-point Likert scale based on the relevancy to the conceptual framework (1 = not relevant, 2 = somewhat relevant, 3 = relevant, 4 = quite relevant, 5 = highly relevant). The ratings were calculated using **Item-content Validity Index (I-CVI)**. For each item, the I-CVI was computed as the number of experts who rated an item on the top two boxes (i.e., quite relevant or highly relevant), divided by the total number of experts. The level of agreement between experts should be as high as possible to make the I-CVI acceptable. The data on content validation is presented in Table 6 under different themes/constructs. These themes/constructs were clustered either as pre-birth themes/constructs or post-birth themes/constructs, sections 1 through 8 cover pre-birth constructs, and sections 9 through 16 cover during and post-birth constructs. The average rating of each section was more than 4 (on a scale of 1 to 5), and the I-CVI is more than 80% for all the sections of the VASA questionnaire. Relatively lower I-CVI was observed for *"symptoms before pregnancy"* (81.43%) and *"Birth History of Child"* (81.82%) sections.

**(c) Discriminant validity (construct validity) assessment.** It shows the distinctiveness of different constructs within the questionnaire and helps to identify whether the items on the questionnaire that are not supposed to be related to each other are unrelated or not. It is one of the types of construct validity. The **"VASA Questionnaire-version-1"** (after the face validation exercise) of the tool was utilized for data collection from the respondents who fall within the inclusion and exclusion criteria. All the mothers identified with a recent history of child mortality, and none of the mothers refused or rejected the interview. Collected data were shared with verbal autopsy reviewers (physicians). The reviewers used their clinical reviewing expertise (a process called *"Physicians certified VA technique-PCVA"*) in assigning the medical cause of death. The complete data (including the CoD assigned by the physicians) were computed on SPSS (version 21) and R for the validation process. For pre-birth constructs, *"birth status of the child"* and for during and post-birth constructs *"diseases or conditions directly leading to*

**Table 6. Average expert rating (from five homologous panels of five different professional expert groups) and overall Item-content Validity Index (I-CVI) by sections of CHERG's VASA questionnaire.**

| VASA questionnaire themes | Content Validity (C.V.) (average rating) | | | | | Average I-CVI (%) |
|---|---|---|---|---|---|---|
| | Rater 1 (n = 5) | Rater 2 (n = 5) | Rater 3 (n = 5) | Rater 4 (n = 5) | Rater 5 (n = 5) | |
| **Content validation (C.V.) of pre-birth themes/constructs* of VASA questionnaire** | | | | | | |
| Demographic and social constructs | 4.56 | 4.44 | 4.47 | 4.53 | 4.53 | 92.94 |
| Symptoms before pregnancy | 4.43 | 4.21 | 4.14 | 4.21 | 4.29 | 81.43 |
| Symptoms during initial two trimesters | 4.45 | 4.24 | 4.27 | 4.24 | 4.36 | 82.42 |
| Symptoms during last trimesters | 4.38 | 4.38 | 4.38 | 4.38 | 4.38 | 87.50 |
| Care Seeking behaviour for symptoms of last trimester | 4.64 | 4.45 | 4.50 | 4.45 | 4.50 | 96.36 |
| Symptoms during labor and care-seeking behaviour | 4.50 | 4.30 | 4.40 | 4.40 | 4.50 | 88.00 |
| Treatment of labor symptoms and post-delivery | 4.34 | 4.16 | 4.19 | 4.22 | 4.28 | 83.75 |
| Knowledge Attitude and Behavior on maternal symptoms | 4.53 | 4.39 | 4.47 | 4.43 | 4.49 | 88.98 |
| **Content validation (C.V.) of during & post-birth themes/constructs* of VASA questionnaire** | | | | | | |
| Birth History of Child | 4.41 | 4.27 | 4.27 | 4.23 | 4.36 | 81.82 |
| Physical Malformations Of Child's Body | 4.57 | 4.43 | 4.36 | 4.43 | 4.50 | 88.57 |
| Child Illness Details | 4.31 | 4.17 | 4.14 | 4.17 | 4.21 | 82.07 |
| Health Seeking Behavior Of Child Illness | 4.63 | 4.50 | 4.56 | 4.56 | 4.56 | 95.00 |
| Health care professionals (HCP) involved during Child Illness | 4.46 | 4.21 | 4.36 | 4.32 | 4.36 | 86.43 |
| Services Cost Burden and Delays | 4.50 | 4.21 | 4.29 | 4.29 | 4.43 | 91.43 |
| Preventive Measures Between Child Birth and Illness | 4.68 | 4.55 | 4.64 | 4.55 | 4.68 | 92.73 |
| Community Efforts during child illness | 4.42 | 4.21 | 4.26 | 4.26 | 4.32 | 85.26 |
| Cause of death-Medical | 4.31 | 4.17 | 4.14 | 4.17 | 4.21 | 82.07 |
| Cause of death-Social | 4.53 | 4.39 | 4.47 | 4.43 | 4.49 | 88.98 |

*Based on a thorough review of the questionnaire, the content of the VASA questionnaire can be divided into two clusters, sections 1 through 8 covers pre-birth constructs and sections 9 through 16 covers during and post-birth constructs. **C.V.**: Content validation; **I-CVI**: Item-content Validity Index;

*death"* are the decisive elements with potential discriminating power. Clear clusters and grouping in response patterns on VASA elements were found for the pre-birth constructs. Therefore, valid response differences between the live birth group and stillbirth group (on the elements of each section) were used for the discriminant validity of the pre-birth constructs of the tool. Similarly, valid response differences among different diseases or conditions directly leading to death were used for the discriminant validity of during or post-birth constructs of the tool.

For <u>pre-birth constructs</u> of the questionnaire, discriminant analysis was performed using birth status as a dependent variable, which was defined in such a way that birth status was coded as "1" (one) in case of stillbirth and "0" (zero) in case of live birth, and elements of each section as discriminating or classification variables. The **goodness of fit summary of discriminant analysis models**, such as eigenvalue (E.V.) of canonical discriminant function, canonical correlation (C.C.), and Wilks' Lambda (W.L.) statistics was used for the discriminant validity measures of the constructs. To analyze the discriminant validity of <u>post-birth constructs</u> of the VASA questionnaire, **multinomial logistic regression** was performed using "disease or condition directly leading to death" as dependent variables and elements of each section of the questionnaire as explanatory variables. Likelihood-Ratio (LR) test and Pseudo R-Square measures (Cox and Snell, Nagelkerke, and McFadden) were used as discriminant validity measures of the constructs.

Out of 100 cases, we had 33 stillbirths and 67 live birth cases. For the analysis, unequal prior probabilities of stillbirth and live birth were assumed and assigned prior probabilities as per

the group size in the data. The deaths included in the study were both, i.e. home-based and institutional (formal and informal health care facilities). The Backward variable elimination method was used to compute the discriminant analysis model. It is vital to comment here at this point that the variables which are not included in the final model do not necessarily mean they are not important, but it means these variable(s) do not provide any additional statistical evidence for the classification of live and stillbirth.

The goodness of fit summary of discriminant analysis models of each of the pre-birth constructs of the VASA questionnaire is presented in Table 7. The **eigenvalue (E.V.)** of canonical discriminant function is *the ratio of the between-groups sum of squares to within groups sum of squares*. A high value of eigenvalue indicates that the canonical function explains sufficiently higher variation in data. Similarly, the **canonical correlation (C.C.)** coefficient shows the predictive ability of the canonical discriminant function. The value of C.C. ranges from 0 to 1, with a value closer to 1 showing the strong predictive ability of canonical discriminant function. The third goodness of fit statistics i.e., **Wilks' Lambda** (ranging from 0 to 1), was calculated to be the proportion of the total variance in the discriminant scores not explained by differences among the groups. A small value of W.L. and statistically significant (p-value of <0.05) shows a good fit of the discriminant model.

The goodness of fit summary of multinomial logistic regression models of each of during & post-birth constructs of the VASA questionnaire is presented in Table 7. Statistically significant **Likelihood-Ratio (LR) test** result infers statistically significant improvement over the

**Table 7. Discriminant validation of CHERG's VASA questionnaire.**

| The goodness of fit summary of discriminant analysis models for 'pre-birth constructs' | | | | | |
|---|---|---|---|---|---|
| VASA questionnaire constructs and sections | Eigenvalue | Canonical Correlation | Wilks' Lambda | | |
| Sections | | | Wilks' Lambda | Chi-square | p-value |
| Demographic and social constructs | 37.44 | 0.987 | 0.026 | 339.37 | <0.001 |
| Symptoms before pregnancy | 2.76 | 0.86 | 0.27 | 127.74 | <0.001 |
| Symptoms during initial two trimesters | 2.76 | 0.86 | 0.27 | 127.74 | <0.001 |
| Symptoms during last trimester | 91.13 | 0.871 | 0.241 | 422.91 | <0.001 |
| Care Seeking behavior for symptoms of last trimester | 1700.51 | 0.615 | 0.621 | 278.97 | <0.001 |
| Symptoms during labor and care seeking behavior | 120.88 | 0.92 | 0.154 | 446.68 | <0.001 |
| Treatment of labor symptoms and post delivery | 1042.16 | 0.993 | 0.013 | 646.35 | <0.001 |
| Knowledge Attitude and Behavior on maternal symptoms | 1475.99 | 0.952 | 0.093 | 682.34 | <0.001 |
| The Goodness of fit summary of multinomial logistic regression models for 'during and post-birth constructs' | | | | | |
| Sections | LR Tests (p-value) | | Cox and Snell | Nagelkerke | McFadden |
| Birth of Child | <0.001 | | 0.968 | 0.996 | 0.963 |
| Physical Malformations of Child's Body | <0.001 | | 0.968 | 0.996 | 0.963 |
| Child Illness Details | <0.001 | | 0.968 | 0.996 | 0.963 |
| Health Seeking Behavior of Child Illness | <0.001 | | 0.968 | 0.996 | 0.963 |
| Health care professionals (HCP) involved during Child Illness | <0.001 | | 0.968 | 0.996 | 0.963 |
| Services Cost Burden and Delays | <0.001 | | 0.956 | 0.983 | 0.87 |
| Preventive Measures between childbirth and illness | <0.001 | | 0.968 | 0.996 | 0.963 |
| Community Efforts during child illness | <0.001 | | 0.956 | 0.983 | 0.87 |
| Cause of death-Biological | <0.001 | | 0.956 | 0.963 | 0.963 |
| Cause of death-Social | <0.001 | | 0.964 | 0.985 | 0.89 |

**C.V.**: Content validation; **D.V.**: Discriminant Validation; **R.A.**: Reliability Analysis; **I-CVI**: Item-content Validity Index; **E.V.**: Eigenvalue; **C.C.**: Canonical correlation; **W.L.**: Wilk's Lambda; **C.A.**: Cronbach's Alpha;

null model. **Pseudo R-Square measures (Cox and Snell, Nagelkerke, and McFadden)** range from 0 to 1, with a value closer to "1" (one) indicating a good model fit.

A high eigenvalue of canonical discriminant function (>1), a high value of canonical correlation (>0.8), and a statistically significant Wilks' Lambda, with a p-value of <0.001, validate the good fit of discriminant models for pre-birth constructs of VASA questionnaire. A high value of Pseudo R-Square measures (>0.8) and a statistically significant LR test, with a p-value <0.001, validate the good fit of multinomial logistic regression models for during & post-birth constructs of the VASA questionnaire. Hence, underlying constructs of the VASA questionnaire, in all sections (Table 7), were found to be distinctive and well-established to address the conceptual framework. More than 85.07% accuracy of the discriminant functions and the multinomial logistic regression model was observed in all sections of the VASA questionnaire.

**(d) Reliability assessment.** Reliability analysis measures the tool's capacity to provide a consistent result. Use of **Cronbach's Alpha** statistics was utilized to measure the reliability of constructs. Due to the dominantly dichotomous nature of data, the calculation of Cronbach's Alpha based on the Pearson correlations matrix would be incorrect or potentially very biased. Therefore, "hetcor" function of "polycor" package of R was used to compute the appropriate heterogeneous correlation matrix. The 'hetcor' function is capable of calculating Pearson correlations (for numeric data), polyserial correlations (for numeric and ordinal data), and polychoric correlations (for ordered or non-ordered factors). Reliability for the different sections of the VASA questionnaire is presented in Table 7. The universal rule of thumb is that, if the value of **Cronbach's alpha** (C.A.) is 0.70 and more, the reliability is good, if the value of C.A. is between 0.80 and 0.90 the reliability is better, and if the value is 0.90 and beyond, the reliability is best. In our study, the Cronbach's Alpha values are above 0.90 across all the constructs of pre-birth, during & post-birth constructs of the VASA questionnaire (except "Demographic and social constructs" and "Care Seeking behaviour for symptoms of last trimester" which are 0.843 and 0.876 respectively) (Table 8).

**Table 8. Reliability analysis of CHERG's VASA questionnaire.**

| VASA questionnaire constructs and section | Specificity (%) | Sensitivity (%) | Accuracy (%) | Cronbach's Alpha |
|---|---|---|---|---|
| **Reliability accuracy of discriminant analysis models for pre-birth constructs** | | | | |
| Demographic and social constructs | 100.00 | 100.00 | 100.00 | 0.843 |
| Symptoms before pregnancy | 100.00 | 72.73 | 91.00 | 0.909 |
| Symptoms during initial two trimesters | 100.00 | 72.73 | 91.00 | 0.944 |
| Symptoms during last trimester | 88.00 | 100.00 | 92.00 | 0.894 |
| Care Seeking behaviour for symptoms of last trimester | 76.00 | 100.00 | 85.00 | 0.876 |
| Symptoms during labor and care-seeking behaviour | 100.00 | 100.00 | 100.00 | 0.954 |
| Treatment of labor symptoms and post delivery | 100.00 | 100.00 | 100.00 | 0.933 |
| Knowledge Attitude and Behavior on maternal symptoms | 100.00 | 100.00 | 100.00 | 0.940 |
| **Reliability and accuracy of multinomial logistic regression models for during and post-birth constructs** | | | | |
| Birth of Child | 100.00 | 100.00 | 100.00 | 0.907 |
| Physical Malformations of Child's Body | 100.00 | 24.20 | 75.00 | 0.947 |
| Child Illness Details | 100.00 | 100.00 | 100.00 | 0.948 |
| Health Seeking Behavior of Child Illness | 100.00 | 100.00 | 100.00 | 0.945 |
| Health care professionals (HCP) involved during Child Illness | 100.00 | 100.00 | 100.00 | 0.928 |
| Services Cost Burden and Delays | 100.00 | 100.00 | 100.00 | 0.95 |
| Preventive Measures between childbirth and illness | 100.00 | 100.00 | 100.00 | 0.973 |
| Community Efforts during child illness | 100.00 | 100.00 | 100.00 | 0.942 |
| Cause of death-Biological | 100.00 | 100.00 | 100.00 | 0.953 |
| Cause of death-Social | 100.00 | 100.00 | 75.00 | 0.973 |

Classification of sensitivity, specificity, and accuracy of the models developed based on the elements of different sections of the VASA questionnaire are presented in Table 8 below. 100% specificity, 100% sensitivity and 100% accuracy of the discriminant functions in classifying births to live and stillbirths were observed in all sections, except, symptoms before pregnancy and symptoms during the initial two-trimester sections of the VASA questionnaire. For which 100%, 72.73% and 91% of specificity, sensitivity, and accuracy were observed, and for these two sections lowest content validity was observed (see Table 8). Cronbach's Alpha values were found higher than 0.80, indicating the reliability and internal consistency of the underlying constructs.

## Discussion

This research validated CHERG's integrated VASA tool using rigorous methodology and substantially all the fundamental methodologies for determining questionnaire validity and reliability. The validation exercise showed good I-CVI (81.43%), Cronbach's Alpha reliability values between 0.843 and 0.973, 81%–100% discriminant accuracy in categorising live and stillbirths, and 97% in assigning medical and social causes of child deaths.

After this current study, this validated tool will be helpful for future researches to collect valid and reliable data pertaining to child mortality determinants operating in Pakistan. Such a tool was highly required to overcome the data shortage-related issues in the country, where the death registration is highly compromised [3, 20, 46]. When it comes to the reasons for having scanty data on death registration, one of the potential bottlenecks is the delayed and compromised notification of child deaths to the National Civil Registration and Vital Statistics Agency (NCRVS). Because the process of notification is time-consuming, there is no evidence of strict compliance among the general population in death notification [46, 47]. There are no financial benefits or assistance being provided to the community. The death registration process in the NCRVS is severely hampered, and this is true at all levels, from the local community to the national level. The existing data is also very limited and with questionable validity and accuracy [3, 20, 46], this is particularly due to no implementation of standardized methods of data collection and reporting for death notification [46, 48]. Utilization and implementation of this validated tool in identifying mortality-related determinants and in assigning medical and social causes of child deaths, especially of those deaths which have been missed from the National Civil Registration and Vital Statistics will surely help us in overcoming data-related issues pertaining to child mortalities.

Validation studies usually encounter daunting challenges. Most of them include recall biases, information biases, small sample size estimation, etc. To overcome such challenges, the methodology involved in the validation studies should be very concrete and categoric, ensuring that the responses from the validated tool should be able to give information with acceptable accuracy. It is highly essential to undertake such relevant validation studies in developing countries so that it will make the validated version of the VASA tool available for conducting VASA investigations.

Literature shows that there are several versions of VA tools developed by multiple agencies and research teams. Literature reflects several attempts at validation studies of different VA tools that have utilized different methodologies to overcome such challenges and identified the precise CoD with much accuracy [30, 31, 49, 50]. The most commonly used VA tool which underwent validation is the World Health Organization's (WHO) VA tool [2, 27].

Having precisely calculated and appropriate sample size, the results of any validation study show the true validity and reliability of the tool. Compared to our VASA validation study, which included 100 deceased under-five cases, all the other VA validation studies undertook

validation on a comparatively larger sample size and implemented only the VA component [27, 28, 32, 33]. We feel that having a smaller sample size is one of the limitations of our validation study. However, our sample size is purely based on general recommendations from Hertzog [39] (this was purely due to the reason that there are no specific guidelines for the sample size of respondents required for pre-survey validation of the VASA tool, hence this study followed the general guidelines from Hertzog guidelines), which suggests that the minimum sample size for any validation study should be 20% of the sample size calculated for the survey the tool will be utilized for. Aggarwal et al. [27] included 313 neonatal deaths; Nausheen et al. [32] undertook validation on 204 neonatal deaths; the sample size in Soofi et al. [33] was 626 neonatal deaths, and Marsha et al. [28] used 137 neonatal deaths.

Conducting interviews after a long delay subsequent to the death event may affect the quality and reliability of data. This is most likely because the respondent needs to recall the events and happenings of different events that occurred before and around the death event. It is highly recommended that VASA interviews should be undertaken somewhere between 01 months till 06 months provided the family has come out of the mourning period [51]. Conducting interviews within the mourning period may cause distress and influence the respondent's disposition and ability to engage in a VASA interview [51]. P Serina et al. [51], suggest that a long recall period may limit the respondent's ability in recalling and recollecting pertinent facts. Events (or symptoms) with extraordinary severity (or implications) remain in the recall for a much more extended period compared to those with mild to moderate severity [51]. Likewise all the other studies [28, 32, 33, 52], our VASA validation study undertook interviews within six (06) weeks after the mortality event. In our sample size, none of the respondents had any difficulty recalling the events.

The capacity of any questionnaire to correctly assign the proper CoD to any mortality with a degree of precision is called accuracy. When dealing with individual-level data, the diagnostic accuracy of the VASA tool is considered to be satisfactory when the specificity and sensitivity of the questionnaire are a minimum of 90% [53]. However, at a much larger, i.e. national level, if the validation attempt of the VASA tool shows sensitivity and specificity of at least 50% and 90% respectively, the tool is usually considered to be having acceptable diagnostic accuracy [53]. This study found an accuracy of between 97% and 100% in assigning medical and social causes of child deaths respectively. In comparison, the accuracy of the WHO's VA tool validated in Aggarwals' study [27] in identifying different neonatal causes of mortalities was found to be up to between 78% to 92% (except for birth asphyxia-16%); however, the kappa statistics was moderate from 0.46 to 0.55. Similarly, in Soofi et al. [33] the WHO VA tool showed a sensitivity of more than 83.5% for diagnosing different causes of neonatal mortality (except congenital malformation, which was 57%), while the specificity of all the major causes of neonatal deaths was found to be more than 90%. Marsha et al. [28] found the sensitivity of the process of assigning the cause of death ranging from 39% (in diagnosing infection) to 90% (in diagnosing causes related to *too early/too small syndrome*) and specificity ranging from 67% (in diagnosing infection) and 99% (in diagnosing Neonatal Tetanus).

## Strengths of our validation study

Our study is first in its kind as it validates the CHERG's integrated VASA tool, which is based on the most holistic conceptual framework, i.e. TPtoSCF (that addresses the barriers and limitations involved in accessing health care services), that have the capacity to record information required to develop policy in preventing under-five death incidents and improving the child survival estimates in developing countries (like Pakistan) where an unknown, but a large

number of child mortality events have known to be missed and their medical causes have not been assigned.

Moreover, this study is important as it attempted a detailed validation of CHERG's integrated VASA tool by incorporating rigorous methodology and utilizing almost all the core methods suggested for identifying the validity and reliability evidence of any questionnaire [54]. The validation exercise yielded high I-CVI (>81.43%), with Cronbach's Alpha reliability statistics being 0.843 for *"Demographic and social constructs"* and highest of 0.973 for *"Preventive Measures between childbirth and illness"*; the accuracy of between 81% (for *"Care seeking behaviour for symptoms of last trimester")* and 100% of the discriminants classifying births to live and stillbirths. The tool showed an accuracy of between 97% and 100% in assigning medical and social causes of child deaths respectively.

Although different VA questionnaires have been validated in Pakistan, however, some of them are specifically designed for the neonatal population [33], while others are for stillbirths [32]. The CHERG's VASA integrated questionnaire encompasses three age groups and has not been validated for the Pakistani population.

An electronic CAPI (Computer Assisted Personal Interview) notebook format for data collection with skip patterns using CSPro software application has also been developed for this validated tool which was utilized for data collection during the Karachi VASA child mortality study. The primary purpose for transforming the lay paper-based form into a notebook application was to minimize the possibility of data collection and data entry errors. No data collection and data entry errors were encountered possibly due to extensive training of the field staff on the data collection and entry.

## Conclusion

Based on our results, we confirm that the CHERG VASA integrated questionnaire is valid, reliable, and relevant to the conceptual frameworks for the Pakistani Population. The validated CHERG VASA tool can be utilized for establishing medical and social causes of Under-five child deaths in Karachi, Pakistan, and is one of the assets for Pakistan's child health policy with the potential to record relevant data and assign causes of child mortality cases occurring in Pakistan. The evidence extracted from the data and information obtained from the validated VASA tool will surely complement healthcare professionals and policymakers in improving the practice and modifying the policy for enhancing the survival of under-five children in Karachi, Pakistan.

## Supporting information

**S1 File. VASA original English questionnaire and translated version.**
(PDF)

**S2 File. VASA translated version questionnaire.**
(PDF)

## Acknowledgments

We would like to acknowledge the key personnel from the relevant communities for assisting the data collection process. The authors would like to express their sincere gratitude to all the officials who provided invaluable assistance in data collection during this research. We extend our deepest appreciation to the healthcare professionals, administrators, and staff who generously shared their time, expertise, and resources with us. We would like to inform that an unreviewed preprint of this manuscript is already available on Research Square at the link

(https://www.researchsquare.com/article/rs-35489/v1) and Europe PMC (https://europepmc.org/article/ppr/ppr191893) with DOI (10.21203/rs.3.rs-35489/v1).

## Author Contributions

**Conceptualization:** Muhammad Bilal Siddiqui.

**Data curation:** Muhammad Bilal Siddiqui.

**Formal analysis:** Muhammad Bilal Siddiqui.

**Investigation:** Muhammad Bilal Siddiqui.

**Methodology:** Muhammad Bilal Siddiqui.

**Project administration:** Muhammad Bilal Siddiqui, Khadijah Abid.

**Resources:** Muhammad Bilal Siddiqui.

**Software:** Muhammad Bilal Siddiqui.

**Supervision:** Muhammad Bilal Siddiqui, Chiu Wan Ng, Wah Yun Low.

**Validation:** Muhammad Bilal Siddiqui.

**Visualization:** Muhammad Bilal Siddiqui.

**Writing – original draft:** Muhammad Bilal Siddiqui.

**Writing – review & editing:** Muhammad Bilal Siddiqui, Chiu Wan Ng, Wah Yun Low, Khadijah Abid.

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
