## [Decision Letter · Decision Letter 0]

25 Nov 2021

PONE-D-21-23446

Validation of CHERG’S Verbal Autopsy-Social Autopsy (VASA) tool for ascertaining determinants and causes of under-five child mortalities in Pakistan.

PLOS ONE

Dear Dr. Siddiqui,

Thank you for submitting your manuscript to PLOS ONE. After careful consideration, we feel that it has merit but does not fully meet PLOS ONE’s publication criteria as it currently stands. Therefore, we invite you to submit a revised version of the manuscript that addresses the points raised during the review process.

A 'Response to Reviewers' letter that responds to each point raised by the academic editor and reviewer(s). You should upload this letter as a separate file labeled 'Response to Reviewers'.A marked-up copy of your manuscript that highlights changes made to the original version. You should upload this as a separate file labeled 'Revised Manuscript with Track Changes'.An unmarked version of your revised paper without tracked changes. You should upload this as a separate file labeled 'Manuscript'.

We look forward to receiving your revised manuscript.

Kind regards,

Prof. Ritesh G. Menezes, M.B.B.S., M.D., Diplomate N.B.

Academic Editor

PLOS ONE

Journal Requirements:

4. Please ensure that you refer to Figure 1 in your text as, if accepted, production will need this reference to link the reader to the figure.

Reviewers' comments:

Reviewer's Responses to Questions

**Comments to the Author**

1. Is the manuscript technically sound, and do the data support the conclusions?

Reviewer #1: Yes

Reviewer #2: Yes

Reviewer #3: Yes

2. Has the statistical analysis been performed appropriately and rigorously? 

Reviewer #1: Yes

Reviewer #2: Yes

Reviewer #3: I Don't Know

3. Have the authors made all data underlying the findings in their manuscript fully available?

Reviewer #1: Yes

Reviewer #2: Yes

Reviewer #3: Yes

4. Is the manuscript presented in an intelligible fashion and written in standard English?

Reviewer #1: Yes

Reviewer #2: Yes

Reviewer #3: Yes

5. Review Comments to the Author

Reviewer #1: The article is an interesting one that further acknowledges the credibility of VASA tools. Kindly note the following observations:

• Line 40 (abstract) : Criteria for classifying countries as high priority needs to be discussed.

• Line 41 (abstract) : The proportion in terms of actual numbers/percentage of under-five mortalities in the countries mentioned can be included here

• Lines 42-43 (abstract) : Basis for stating that “an unknown but major proportion of such mortalities…” needs to be mentioned.

• Line 97 : Proper definition and differentiation between verbal and social autopsy needs to be highlighted. In addition, the impression given by the sentences is that they are useful primarily in low resource settings whereas the utility of verbal and social autopsies is multi-contextual.

• Line 125 : Seems like a redundant statement which does not add anything important to the paper.

• Lines 131-132 : The authors need to explain the difference between the CHERG’s SA tool and CHERG’s integrated VASA tool. Or at least state the basis for the difference.

• Lines 138-140 : Kindly reframe the sentence as it is difficult to understand why this VASA integrated tool study is one of a kind.

• Lines 147-149 : It is not clear whether Karachi’s VASA child mortality study and Karachi’s VASA study (VASA survey) are the same or different entities.

• Lines 152-155 : It would be helpful if the authors explained why they went in for general guidelines rather than VASA specific guidelines while determining the sample size for this validation study.

• Line 171 : The location of the Advance Educational Institute and Research Centre needs to be clarified. Is it based in Pakistan? Or is it a Malaysian/Canadian based centre. The authors also need to explain why local ethical clearance was not needed/not sought for a study involving Pakistani citizens carried out in Pakistan.

• Line 176 : It would give the study better credence if all the “many more” subject specialists were mentioned fully.

• Lines 179-179 : the methodology of Physician certified verbal autopsy CoD and Algorithm Certified Verbal Autopsy CoD should be expanded on.

• Lines 365-366 : The statements pertain only to the WHO VA tool without a comparison with the CHERG VA tool. It needs to be specified if these are separate tools. Though it has been mentioned that the CHERG tools are the result of collaboration between the WHO and UNICEF, the reasons for development of a child centric VA tool needs to be brought in.

• Lines 370-373 : As mentioned earlier, it would be helpful if the authors explained why they went in for Hertzog guidelines rather than VASA specific guidelines while determining the sample size for this validation study.

• Lines 415-418 : The sentence is difficult to comprehend. Kindly look into simplifying the sentence/breaking it up so that the reader can better understand what the authors are trying to convey.

• Lines 424-426 : Some of the reasons for the compromised birth and death registration can be mentioned as well. Doing so will further stress on the need for using VASA tools.

• Lines 426-427 : As above, the reasons for the questionable accuracy and validity of the data can be added here.

• Lines 454-456 : The authors have mentioned that a CAPI notebook format has been developed to minimize data entry errors. It is suggested that the authors elaborate on the frequency of such errors using the paper-based form and whether there was a possibility of their study also having such data entry errors.

• Lines 461-464 : The authors mention a need to have “specific recommendations for the estimated sample size explicitly…..”. However, earlier in the article they have stated that “literature shows very succinct specific guidelines on the sample size OF RESPONDENTS required for pre-survey validation exercise of VASA tool” (lines 152-153). This is a contradiction.

Reviewer #2: Dear Authors,

Line 290: The authors needs to clarify, whether the still birth / live birth mentioned in the study , where due to Institutional delivery with proper health care assistance or home delivery by dhais.

Reviewer #3: In this study by Siddiqui et al, authors validate the Verbal Autopsy and Social Autopsy (VASA) tools in 100 mothers of recently deceased under-5 children in Karachi, Pakistan. Authors have done a commendable job in presenting this data. The manuscript is overall well-written. However, I have a few suggestions:

Major Comments

Introduction

1) It would be helpful if authors provide some statistics pertaining the number of under 5-mortality rates in few lower middle-income countries, particularly Pakistan. This is important, as it stresses and contextualizes statistically the extremely high number of these premature deaths, which builds a stronger case for the use of these tools.

Methods

2) Which slums from Karachi were selected? Important mention the names for complete transparency as this links directly to their socio-economic class.

3) Table 1: in VA-6, any examples of the open-ended questions which were asked?

Results

4) Results should provide more background details. How many women were approached who refused to answer the questionnaire? Reasons for refusal? (important to state this was not because they found the questionnaire tough to interpret)

5) Some more details regarding face validation should be added. Of the 100 respondents, did all of them understand every question and option clearly? If not, adding a table might make this more clear

6) A good chunk of results includes details regarding how content validation/discrimination etc was done and the details regarding statistical analysis. Would suggest authors to have a separate subsection of statistical methods at the end of Methods where they can describe all the statistical models used. In results, the main focus should be the outcome of these analyses which are mentioned very briefly right now. I would suggest authors to also mention in text the key points from tables, especially for Table 4.

Discussion

7) Line 426-427: Authors should comment and compare the results of VASA validity observed in their analysis with that existing in literature.

8) This study has many limitations which must be acknowledged in a separate paragraph. First, data was collected from one city only in a limited number of people. Second, did all the respondents belong to low socio-economic class? Third, since all interviews were conducted, interview bias can play a pivotal role.

Minor

Line 54: “one hundred randomly selected mothers of deceased , with a recent child death event” would suggest authors to revise this line to amend the grammatical error

Line 152: Would request the authors to review why ‘OF RESPONDENTS’ is capitalized?

6. PLOS authors have the option to publish the peer review history of their article (what does this mean?). If published, this will include your full peer review and any attached files.

Reviewer #1: No

Reviewer #2: **Yes: **Dr Jagadish Rao Padubidri

Reviewer #3: No

---

## [Author Response · Author response to Decision Letter 0]

22 Feb 2022

Line 40 (abstract): Criteria for classifying countries as high priority needs to be discussed. Deleted and modified as per the suggestion. The introduction section of abstract has been modified as per the suggestion.

Line 41 (abstract): The proportion in terms of actual numbers/percentage of under-five mortalities in the countries mentioned can be included here. Deleted and modified as per the suggestion. The introduction section of abstract has been modified as per the suggestion.

Lines 42-43 (abstract): Basis for stating that “an unknown but major proportion of such mortalities…” needs to be mentioned. Explained. The introduction section of abstract has been modified as per the suggestion.

Line 97 : Proper definition and differentiation between verbal and social autopsy needs to be highlighted. In addition, the impression given by the sentences is that they are useful primarily in low resource settings whereas the utility of verbal and social autopsies is multi-contextual. Mentioned. Lines no 55-63. 

Line 125 : Seems like a redundant statement which does not add anything important to the paper. Deleted. Line 94.

Lines 131-132 : The authors need to explain the difference between the CHERG’s SA tool and CHERG’s integrated VASA tool. Or at least state the basis for the difference. Deleted the mention of CHERG’s SA tool, as it caused a confusion in understanding. Lines 99-100.

Lines 138-140 : Kindly reframe the sentence as it is difficult to understand why this VASA integrated tool study is one of a kind. Reframed. Lines 105-107. 

Lines 147-149 : It is not clear whether Karachi’s VASA child mortality study and Karachi’s VASA study (VASA survey) are the same or different entities. 

Lines 152-155 : It would be helpful if the authors explained why they went in for general guidelines rather than VASA specific guidelines while determining the sample size for this validation study. Explained. Lines 120-127. 

Line 171 : The location of the Advance Educational Institute and Research Centre needs to be clarified. Is it based in Pakistan? Or is it a Malaysian/Canadian based centre. The authors also need to explain why local ethical clearance was not needed/not sought for a study involving Pakistani citizens carried out in Pakistan. Explained. This paper is in continuation of a Phd study conducted by the corresponding author (M. Bilal Siddiqui) who is a student in University of Malaya, hence the university’s ethical clearance was sought. Moreover, the study was conducted in Karachi, Pakistan, the local (Karachi, Pakistan’s) ethical clearance was also sought by AEIRC. Lines 153-157. 

Line 176 : It would give the study better credence if all the “many more” subject specialists were mentioned fully. Removed the phrase ‘many more’ lines 211-212, as none of the specialists other than the mentioned ones were involved. It was a typing and writing error.

Lines 179-179 : the methodology of Physician certified verbal autopsy CoD and Algorithm Certified Verbal Autopsy CoD should be expanded on. It is again a writing error. The Algorithm Certified Verbal Autopsy CoD was not utilized. Only, the Physician certified verbal autopsy CoD was performed. Corrected and explained the Physician certified verbal autopsy CoD. Lines 212-217.

Lines 365-366 : The statements pertain only to the WHO VA tool without a comparison with the CHERG VA tool. It needs to be specified if these are separate tools. Though it has been mentioned that the CHERG tools are the result of collaboration between the WHO and UNICEF, the reasons for development of a child centric VA tool needs to be brought in. This is a good idea for having a child centric VA tool, rather than different versions of VA tools. In the manuscript, a statement has been added that the WHO VA tool is different from the CHERG VA tool. Lines 374-378. 

Lines 370-373 : As mentioned earlier, it would be helpful if the authors explained why they went in for Hertzog guidelines rather than VASA specific guidelines while determining the sample size for this validation study.In the manuscript it was initially mentioned that there are succinct specific guidelines on sample size of respondents required for pre-survey validation exercise of VASA tool, however, it was a mistakenly written statement, and actually there are no specific guidelines for it, hence because of this the authors have followed the general guidelines from Hertzog guidelines in the manuscript. The statement has been updated now. Lines 120-123, 381-392.

Lines 415-418 : The sentence is difficult to comprehend. Kindly look into simplifying the sentence/breaking it up so that the reader can better understand what the authors are trying to convey. Corrected. Lines 433-435. 

Lines 424-426 : Some of the reasons for the compromised birth and death registration can be mentioned as well. Doing so will further stress on the need for using VASA tools. Added. Lines 441-449.

Lines 426-427: As above, the reasons for the questionable accuracy and validity of the data can be added here. Modified. Lines 449-452. 

Lines 454-456: The authors have mentioned that a CAPI notebook format has been developed to minimize data entry errors. It is suggested that the authors elaborate on the frequency of such errors using the paper-based form and whether there was a possibility of their study also having such data entry errors. This validation study did not encounter any errors between the CAPI and PAPI platforms, however, no inconsistencies were encountered between the PAPI and CAPI platforms. The only difference between the two platforms was ease of data collection and less time utilized for data collection by the data collectors. Lines 480-484. 

Lines 461-464 : The authors mention a need to have “specific recommendations for the estimated sample size explicitly…..”. However, earlier in the article they have stated that “literature shows very succinct specific guidelines on the sample size OF RESPONDENTS required for pre-survey validation exercise of VASA tool” (lines 152-153). This is a contradiction. The manuscript still recommends that there is a need to have “specific recommendations for the estimated sample size explicitly required for the validation process of the VASA tool. however, earlier in the manuscript, it was a mistakenly written statement that “literature shows very succinct specific guidelines on the sample size OF RESPONDENTS required for pre-survey validation exercise of VASA tool” (lines 120-123, 381-392.). Actually, there are no specific guidelines for it, hence because of this the authors have followed the general guidelines from Hertzog guidelines in the manuscript. These statements has been updated now where ever it was mentioned such in the manuscript.

Line 290: The authors needs to clarify, whether the still birth / live birth mentioned in the study, where due to Institutional delivery with proper health care assistance or home delivery by dhais. Added. Lines 301-302.

Introduction

1) It would be helpful if authors provide some statistics pertaining the number of under 5-mortality rates in few lower middle-income countries, particularly Pakistan. This is important, as it stresses and contextualizes statistically the extremely high number of these premature deaths, which builds a stronger case for the use of these tools. The overall text in the introduction has been modified. Lines 21-23. 

Methods

2) Which slums from Karachi were selected? Important mention the names for complete transparency as this links directly to their socio-economic class. Mentioned as suggested. Lines 128-136.

Methods

3) Table 1: in VA-6, any examples of the open-ended questions which were asked? Mentioned in the table itself. Table-2, Line 184. 

Results

4) Results should provide more background details. How many women were approached who refused to answer the questionnaire? Reasons for refusal? (important to state this was not because they found the questionnaire tough to interpret). Mentioned in the section of ‘Discriminant validity (construct validity) assessment. Lines 268-335.

Results

5) Some more details regarding face validation should be added. Of the 100 respondents, did all of them understand every question and option clearly? If not, adding a table might make this more clear. the face validation was undertaken by the panel of experts and not by the respondents from the deceased family. Lines 236-243.

Results

6) A good chunk of results includes details regarding how content validation/discrimination etc was done and the details regarding statistical analysis. Would suggest authors to have a separate subsection of statistical methods at the end of Methods where they can describe all the statistical models used. In results, the main focus should be the outcome of these analyses which are mentioned very briefly right now. I would suggest authors to also mention in text the key points from tables, especially for Table 4. the method adopted for calculating the I CVI index through content validation and discriminant validation is explicitly mentioned in the section of content validation. Both of these validation processes included statistical analysis using statistical analysis software. Lines (247-265)(271-322). The explanation of the analysis is already mentioned in details in separate sections of content (247-265) and discriminant (268-335) validity. As per your suggestion, if the explanation of analysis is given as a separate section after the methods section, I feel that the reader might get confused with presentation of explanation. However, if you strongly feel that it should be done, then kindly re-suggest in your review comments, so that I work on it. 

Discussion

7) Line 426-427: Authors should comment and compare the results of VASA validity observed in their analysis with that existing in literature. There is no prior study on the validation of VASA tool, hence the discussion did not include the comparison, however, the VA and SA results were compared with the most relevant studies. Lines 432-433.

Discussion

8) This study has many limitations which must be acknowledged in a separate paragraph. First, data was collected from one city only in a limited number of people. Second, did all the respondents belong to low socio-economic class? Third, since all interviews were conducted, interview bias can play a pivotal role. The issue of generalizability was already addressed while planning the sampling methodology of this validation study. The sampling methodology was not mentioned in the manuscript, now it has been explained in detail. Lines 119-144. 

There was a mixture of respondents from all the wealth quintiles. Please see Table-4. Line 233. 

All the interviews were conducted from a random sample. Line 32 and line 128. 

Minor

Line 54: “one hundred randomly selected mothers of deceased , with a recent child death event” would suggest authors to revise this line to amend the grammatical error. Corrected. Lines 32-33. 

Minor

Line 152: Would request the authors to review why ‘OF RESPONDENTS’ is capitalized? Typographical error. It has been corrected now. Line 120.

---

## [Decision Letter · Decision Letter 1]

31 Mar 2022

PONE-D-21-23446R1

Validation of CHERG’S Verbal Autopsy-Social Autopsy (VASA) tool for ascertaining determinants and causes of under-five child mortalities in Pakistan.

PLOS ONE

Dear Dr. Siddiqui,

Thank you for submitting your manuscript to PLOS ONE. After careful consideration, we feel that it has merit but does not fully meet PLOS ONE’s publication criteria as it currently stands. Therefore, we invite you to submit a revised version of the manuscript that addresses the points raised during the review process.

A 'Response to Reviewers' letter that responds to each point raised by the academic editor and reviewer(s). You should upload this letter as a separate file labeled 'Response to Reviewers'.A marked-up copy of your manuscript that highlights changes made to the original version. You should upload this as a separate file labeled 'Revised Manuscript with Track Changes'.An unmarked version of your revised paper without tracked changes. You should upload this as a separate file labeled 'Manuscript'.

We look forward to receiving your revised manuscript.

Kind regards,

Prof. Ritesh G. Menezes, M.B.B.S., M.D., Diplomate N.B.

Academic Editor

PLOS ONE

Journal Requirements:

Reviewers' comments:

Reviewer's Responses to Questions

**Comments to the Author**

1. If the authors have adequately addressed your comments raised in a previous round of review and you feel that this manuscript is now acceptable for publication, you may indicate that here to bypass the “Comments to the Author” section, enter your conflict of interest statement in the “Confidential to Editor” section, and submit your "Accept" recommendation.

Reviewer #1: All comments have been addressed

Reviewer #4: All comments have been addressed

Reviewer #5: (No Response)

2. Is the manuscript technically sound, and do the data support the conclusions?

Reviewer #1: (No Response)

Reviewer #4: Yes

Reviewer #5: Partly

3. Has the statistical analysis been performed appropriately and rigorously? 

Reviewer #1: (No Response)

Reviewer #4: Yes

Reviewer #5: I Don't Know

4. Have the authors made all data underlying the findings in their manuscript fully available?

Reviewer #1: (No Response)

Reviewer #4: Yes

Reviewer #5: Yes

5. Is the manuscript presented in an intelligible fashion and written in standard English?

Reviewer #1: (No Response)

Reviewer #4: Yes

Reviewer #5: Yes

6. Review Comments to the Author

Reviewer #1: (No Response)

Reviewer #4: Line28,66,68: the term mortalities is not appropriate – needs to be corrected as mortality – including the title

Line 210:what was the educational background of the data collectors?

Table1: the column for frequency can only depict n. denominator of 100 from each cell needs to be removed. The percent column can mention (%) in column heading & the % symbol from each cell can be removed.

All Tables: The percent column can mention (%) in column heading & the % symbol from each cell can be removed.

Table 1:please provide details about unimproved & improved facility mentioned under access to toilets

Table 7: cause of death biological – sensitivity % is missing

Methods section could include details about the characteristics of the deceased children. Age & gender, cause of death, hospitalization, institutional delivery, antenatal history, hospital records if available etc

Line 443:avoid duplication of word specifically

Line 467: it is preferable to avoid superlatives like highly important

Line 484-485: the term error is missing from the statement

Line 490-492: the recommendation is not well suited, considering the present study objectives & hence needs to be removed.

Reviewer #5: Sample Size:

What was the data source for identification of mothers who had lost their Under five children at each of the 12 sites listed? Authors suggest that death registration is far from complete in Karachi city yet they claim mothers were randomly included based on Proportional uota sampling. How was this done? Please clarify?

Line 131 135 : "These 12 locations throughout the city of Karachi were chosen based on criteria to identify geographical points throughout the entire city where residents belonging to the most common and frequently found ethnic backgrounds (Punjabi, Sindhi, Balochi, Siraiki, Pathan, Urdu speaking, Kashmiri, Hindko,and Brahvi) live within at least 2 kilometres of residential locations. This criterion, ensured that the sample is representative of the general population." Unclear? What do the authors mean by living within 2 kms of residential locations? Please rephrase.

Line 170: Authors describe CHERG VASA tool as a uantitative tool. This is not so. It oncludes open ended text fields as well.

Line 216-218: Authors have not elaborated details of PCVA process. Did a single physician review a particular VA and assign most probable CoD? If so it introduces a subjective bias? Did more than one physician independently review each VA form? If so what was the agreement between physicians and how were disagreements adjudicated? How were the responses made available to the physicians? It is well known that in PCVA CoD is largely assigned based on the text narrative rather than the close ended responses. What measures were taken to minimize this phenomenon?

In the PAPI what was the completion rate for the entire questionnaire after accounting for skip patterns?

Section V5.12 - V5.16 of the CHERG VASA questionnaire record CoD listed on the death certificate. Was this information masked from the Physicians? If not it calls into question the very high values of 97% accuracy for CoD assigned based on VA and may invalidate the findings of the entire study. Please give details

7. PLOS authors have the option to publish the peer review history of their article (what does this mean?). If published, this will include your full peer review and any attached files.

Reviewer #1: No

Reviewer #4: No

Reviewer #5: **Yes: **Ayon Gupta

---

## [Author Response · Author response to Decision Letter 1]

6 Sep 2022

AUTHORS COMMENTS AUTHOR’S COMPLIANCE/COMMENTS

Reviewer #4: Line28,66,68: the term mortalities is not appropriate – needs to be corrected as mortality – including the title (Done, see title, please see whole document. I have amended your suggestion throughout the document)

Line 210: what was the educational background of the data collectors (Done. please see line 406-407)

Table1: the column for frequency can only depict n. denominator of 100 from each cell needs to be removed. The percent column can mention (%) in column heading & the % symbol from each cell can be removed. (Done, please see all tables)

All Tables: The percent column can mention (%) in column heading & the % symbol from each cell can be removed. (Done, please see all tables)

Table 1:please provide details about unimproved & improved facility mentioned under access to toilets (described in the table 1 at the bottom of the respective table. Also referenced).

Table 7: cause of death biological – sensitivity % is missing (Done, please see table no-08)

Methods section could include details about the characteristics of the deceased children. Age & gender, cause of death, hospitalization, institutional delivery, antenatal history, hospital records if available etc (Done in table 5)

Line 443:avoid duplication of word specifically (Done)

Line 467: it is preferable to avoid superlatives like highly important (Done)

Line 484-485: the term error is missing from the statement (Done, please see Line no 1351)

Line 490-492: the recommendation is not well suited, considering the present study objectives & hence needs to be removed. (Deleted)

Reviewer #5: Sample Size:

What was the data source for identification of mothers who had lost their Under five children at each of the 12 sites listed? Authors suggest that death registration is far from complete in Karachi city (deleted from the manuscript as the statement was confusing) yet they claim mothers were randomly included based on Proportional uota sampling. How was this done? Please clarify? (Done. clarified between lines 225-257) (deleted from the manuscript as the statement was confusing)

(Done. clarified between lines 225-257)

Line 131 135 : "These 12 locations throughout the city of Karachi were chosen based on criteria to identify geographical points throughout the entire city where residents belonging to the most common and frequently found ethnic backgrounds (Punjabi, Sindhi, Balochi, Siraiki, Pathan, Urdu speaking, Kashmiri, Hindko,and Brahvi) live within at least 2 kilometres of residential locations. This criterion, ensured that the sample is representative of the general population." Unclear? What do the authors mean by living within 2 kms of residential locations? Please rephrase. (typo error. Rewritten. Done, please see Line 225-252)

Line 170: Authors describe CHERG VASA tool as a uantitative tool. This is not so. It includes open ended text fields as well. (Done, please see Line 355-357)

Line 216-218: Authors have not elaborated details of PCVA process. Did a single physician review a particular VA and assign most probable CoD? If so it introduces a subjective bias? Did more than one physician independently review each VA form? If so what was the agreement between physicians and how were disagreements adjudicated? How were the responses made available to the physicians? It is well known that in PCVA CoD is largely assigned based on the text narrative rather than the close ended responses. What measures were taken to minimize this phenomenon? (explained between lines 412-419).

In the PAPI what was the completion rate for the entire questionnaire after accounting for skip patterns? (Done, please see Line no 401-403)

Section V5.12 - V5.16 of the CHERG VASA questionnaire record CoD listed on the death certificate. Was this information masked from the Physicians? If not it calls into question the very high values of 97% accuracy for CoD assigned based on VA and may invalidate the findings of the entire study. Please give details (Done, Line no: 410-431)

---

## [Decision Letter · Decision Letter 2]

10 Oct 2022

PONE-D-21-23446R2Validation of CHERG’S Verbal Autopsy-Social Autopsy (VASA) tool for ascertaining determinants and causes of under-five child mortalities in Pakistan.

PLOS ONE

Dear Dr. Siddiqui,

Thank you for submitting your manuscript to PLOS ONE. After careful consideration, we feel that it has merit but does not fully meet PLOS ONE’s publication criteria as it currently stands. Therefore, we invite you to submit a revised version of the manuscript that addresses the points raised during the review process.

Please submit your revised manuscript by Nov 24 2022 11:59PM. Please include the following items when submitting your revised manuscript:

A 'Response to Reviewers' letter that responds to each point raised by the academic editor and reviewer(s). You should upload this letter as a separate file labeled 'Response to Reviewers'.A marked-up copy of your manuscript that highlights changes made to the original version. You should upload this as a separate file labeled 'Revised Manuscript with Track Changes'.An unmarked version of your revised paper without tracked changes. You should upload this as a separate file labeled 'Manuscript'.

We look forward to receiving your revised manuscript.

Kind regards,

Prof. Ritesh G. Menezes, M.B.B.S., M.D., Diplomate N.B.

Academic Editor

PLOS ONE

Journal Requirements:

Please review your reference list to ensure that it is complete and correct. If you have cited papers that have been retracted, please include the rationale for doing so in the manuscript text or remove these references and replace them with relevant current references. Any changes to the reference list should be mentioned in the rebuttal letter that accompanies your revised manuscript. If you need to cite a retracted article, indicate the article’s retracted status in the References list and also include a citation and full reference for the retraction notice.

Additional Editor Comments:

The list of authors (manuscript submitted to PLOS ONE) is different from the list of authors in the preprint version of the same manuscript published on Research Square. PLOS ONE follows the COPE guidelines for changes in authorship. If this ethical aspect related to authorship is not addressed appropriately by the corresponding author, then the manuscript will not be considered for publication in PLOS ONE. Besides, the manuscript should be edited for minor language corrections and reviewers' comments should be addressed.

Reviewers' comments:

Reviewer's Responses to Questions

**Comments to the Author**

1. If the authors have adequately addressed your comments raised in a previous round of review and you feel that this manuscript is now acceptable for publication, you may indicate that here to bypass the “Comments to the Author” section, enter your conflict of interest statement in the “Confidential to Editor” section, and submit your "Accept" recommendation.

Reviewer #5: (No Response)

Reviewer #6: All comments have been addressed

2. Is the manuscript technically sound, and do the data support the conclusions?

Reviewer #5: Yes

Reviewer #6: Yes

3. Has the statistical analysis been performed appropriately and rigorously? 

Reviewer #5: Yes

Reviewer #6: Yes

4. Have the authors made all data underlying the findings in their manuscript fully available?

Reviewer #5: Yes

Reviewer #6: Yes

5. Is the manuscript presented in an intelligible fashion and written in standard English?

Reviewer #5: Yes

Reviewer #6: Yes

6. Review Comments to the Author

Reviewer #5: Authors have largely addressed issues raised by this reviewer in the earier review. However certain suggestions have been made. Please see reviewer comments attached as notes in the uploaded reviewed manuscript. Request address the same

Reviewer #6: I have few comments that can be incorporated into the paper:

1. Full form of I-CVI should be mentioned in abstract

2. Line 405-413: The author can change this part of the manuscript and add the points to introduction and should begin their study by the summary of their results.

3. Line 413-415: A brief explanation of how their study ensures availability of the tool, can be incorporated in the manuscript.

7. PLOS authors have the option to publish the peer review history of their article (what does this mean?). If published, this will include your full peer review and any attached files.

Reviewer #5: **Yes: **Ayon Gupta

Reviewer #6: No

---

## [Author Response · Author response to Decision Letter 2]

16 Oct 2022

The editor commented on the references and updating the references. 

COMMENT: “Please review your reference list to ensure that it is complete and correct. If you have cited papers that have been retracted, please include the rationale for doing so in the manuscript text or remove these references and replace them with relevant current references. Any changes to the reference list should be mentioned in the rebuttal letter that accompanies your revised manuscript. If you need to cite a retracted article, indicate the article’s retracted status in the References list and also include a citation and full reference for the retraction notice.”

Author’s response: The reference list was updated. On line number 74 in the manuscript prior to the current revision and line no 108 in the manuscript submitted after the current R2 review comments, there were some references 4,6–9 was corrected as these were typing error. There errors have been deleted. 

Moreover, between lines 71 and 82 in the manuscript prior to the current revision, there were some references which were mistakenly repeated. Hence these doubled references were removed, which can be verified between the line no 100-109 in the manuscript submitted after the current R2 review comments.

ADDITIONAL EDITOR COMMENTS: On deleting the name of one author, Mr Hassan Ahmed from the manuscript.

Author response: (An email has been sent to Editor Plos One for explaining the rationale of removing Mr Hassan Ahmed from the authors list). The editor has replied back to me on 13th October 2021 saying that the editor has contacted the author for clarification, and I may contact the editor back if I did not hear from editor in the next one week.

REVIEWER #5 COMMENTS: Authors have largely addressed issues raised by this reviewer in the earlier review. However certain suggestions have been made. Please see reviewer comments attached as notes in the uploaded reviewed manuscript. Request address the same.

Author’s comments: 

1. Table-1: Check table heading and column content: I have deleted “X (SAMPLE SIZE)”

2. Line 169 in the manuscript prior to the current R2 revision: I have replaced “counted” to “included”. See line no 339 in the manuscript after the current R2 review comments.

3. Line 186 in the manuscript prior to the current R2 revision: I have replaced “quantitative” to “semi-quantitative”. See line no 356 in the manuscript submitted after the current R2 review comments. 

4. Line 203 in the manuscript prior to the current R2 revision: check grammar. The grammar of the sentence starting from Line no 201-203 (in the manuscript submitted after the current R2 review comments) has been improved. 

REVIEWER #6 COMMENTS: I have few comments that can be incorporated into the paper:

1. Full form of I-CVI should be mentioned in abstract. 

2. Line 405-413: The author can change this part of the manuscript and add the points to introduction and should begin their study by the summary of their results.

3. Line 413-415: A brief explanation of how their study ensures availability of the tool, can be incorporated in the manuscript.

Author’s response: 

1. Done in the abstract. See line no 54 in the manuscript submitted after the current R2 review comments.

2. Line 405-413 The contents were deleted from discussion. Such details were already available in Introduction, so to avoid duplicacy, the deleted points were not added in Introduction. The discussion was initiated with the summary of the results of this validation study. 

3. Line 413-415: The line was rephrased into much better context and avoided the concept of “ensuring the availability of the tool”.

ADDITIONAL AUTHORS’ SELF CHANGES: 

1. I have added my personal email address (drbilals@gmail.com), which in case if I am not accessible on the initial two email addresses, can be use for contacting me. If it is not acceptable by the journal, you have complete rights to not to include in the manuscript.

2. In the abstract line no 30-31 in the manuscript submitted after the current R2 review comments, the term CHERG was explained with a full form as “Child Health Epidemiology Reference Group”. I felt that it might have been missed from reviewers, hence I corrected it from my end. If it is not acceptable by the journal, you have complete rights to not to include in the manuscript.

---

## [Editor Report · Decision Letter 3]

26 Oct 2022

PONE-D-21-23446R3

Validation of CHERG’S Verbal Autopsy-Social Autopsy (VASA) tool for ascertaining determinants and causes of under-five child mortality in Pakistan.

PLOS ONE

Dear Dr. Siddiqui,

Thank you for submitting your manuscript to PLOS ONE. The revised manuscript is provisionally accepted for publication, pending ethical issues related to authorship change.

Please submit the latest version of the revised manuscript by Dec 10 2022 11:59PM. When you're ready to submit your revision, log on to https://www.editorialmanager.com/pone/ and select 'Submissions Needing Revision' to proceed. While doing so, you must include all relevant files including the 'Request for Change to Authorship Form' signed by all those concerned. For further details, refer to the 'Additional Editor Comments' mentioned below.

A 'Response to Reviewers' letter that responds to each point raised by the editor and reviewer(s). You should upload this letter as a separate file labeled 'Response to Reviewers'.A marked-up copy of your manuscript that highlights changes made to the original version. You should upload this as a separate file labeled 'Revised Manuscript with Track Changes'.An unmarked version of your revised paper without tracked changes. You should upload this as a separate file labeled 'Manuscript'.

We look forward to receiving the latest version of your revised manuscript.

Kind regards,

Prof. Ritesh G. Menezes, M.B.B.S., M.D., Diplomate N.B.

Academic Editor

PLOS ONE

Journal Requirements:

Please review your reference list to ensure that it is complete and correct. If you have cited papers that have been retracted, please include the rationale for doing so in the manuscript text or remove these references and replace them with relevant current references. Any changes to the reference list should be mentioned in the rebuttal letter that accompanies your revised manuscript. If you need to cite a retracted article, indicate the article’s retracted status in the References list and also include a citation and full reference for the retraction notice.

Additional Editor Comments:

The list of authors (manuscript submitted to PLOS ONE) differs from the list of authors in the preprint version of the same manuscript published on Research Square. PLOS ONE follows the COPE guidelines for changes in authorship. The present reply provided by the corresponding author is not acceptable. If this ethical aspect related to authorship change is not addressed appropriately by the corresponding author, then the manuscript will not be considered for publication in PLOS ONE. This is the final opportunity provided to the corresponding author to address this issue satisfactorily.

plos-one-change-to-authorship-form.docx (live.com)

https://view.officeapps.live.com/op/view.aspx?src=https%3A%2F%2Fstorage.googleapis.com%2Fplos-published-prod%2F13d0%2Fplos-one-change-to-authorship-form.docx%3FX-Goog-Algorithm%3DGOOG4-RSA-SHA256%26X-Goog-Credential%3Dwombat-sa%2540plos-prod.iam.gserviceaccount.com%252F20221025%252Fauto%252Fstorage%252Fgoog4_request%26X-Goog-Date%3D20221025T141724Z%26X-Goog-Expires%3D86400%26X-Goog-SignedHeaders%3Dhost%26X-Goog-Signature%3D776f36e892850cbe666e205450d9b799af8031c3d6682d9ecff348d9fb448c438fa7644712b44a47bbd16f986a41e697956b2481c463ae000ac78e6f7a5be9a080193a9768deb8e57dc9df6c32b04e88a56c3a936c1f5f0d09eefae7c3d396d96c9d17cda529ab61d110da727e387754bd64051097e28b2eb548a72c006e382240a02a7b6bda4520bddbc57c3988d9412911f10d7d588f8ef6cb178cee857e3c152b38bfdc5b85b86ea7d08089a77e164bd121ce7a8af15c8d8d5247d6cf819af3e92c81c7aea0365bcb4b9d4eb1fa734f416cd29183f14ba6e6aff4666124287d7466a5072740d5725b6c070ef52dc02c301db990623364398d02c29ccc5b0c&wdOrigin=BROWSELINK

Refer to the link above and complete the “Request for Change to Authorship Form”. IN ADDITION, PLEASE NOTE THAT THIS FORM MUST BE SIGNED BY ALL THE AUTHORS (INCLUDING CURRENT CO-AUTHORS, THOSE TO BE ADDED, AND THOSE TO BE REMOVED). ALL THE SIGNATURES MUST BE PRESENT ON THE SAME COPY OF THE FORM WITH THE DATE ON WHICH THE FORM WAS SIGNED BY THE RESPECTIVE AUTHORS.

---

## [Author Response · Author response to Decision Letter 3]

9 Nov 2022

I have attached the Request for change to authorship form for the process to complete.

---

## [Editor Report · Decision Letter 4]

11 Nov 2022

Validation of CHERG’S Verbal Autopsy-Social Autopsy (VASA) tool for ascertaining determinants and causes of under-five child mortality in Pakistan.

PONE-D-21-23446R4

Dear Dr. Ng,

We’re pleased to inform you that your manuscript has been judged scientifically suitable for publication and will be formally accepted for publication once it meets all outstanding technical requirements. Please note that you must follow the reference style as per PLOS ONE guidelines.

Kind regards,

Prof. Ritesh G. Menezes, M.B.B.S., M.D., Diplomate N.B.

Academic Editor

PLOS ONE

Additional Editor Comments:

PLOS ONE submission guidelines should be rechecked by the authors for technical corrections (for instance, reference style) at the time of proof corrections. If another round of revision is required before the stage of proof corrections, the corresponding author is requested to write to the journal.

---

## [Editor Report · Acceptance letter]

26 Oct 2023

PONE-D-21-23446R4 

Validation of CHERG’S Verbal Autopsy-Social Autopsy (VASA) tool for ascertaining determinants and causes of under-five child mortality in Pakistan. 

Dear Dr. Ng:

I'm pleased to inform you that your manuscript has been deemed suitable for publication in PLOS ONE. Congratulations! Your manuscript is now with our production department. 

Kind regards, 

on behalf of

Professor Ritesh G. Menezes 

Academic Editor

PLOS ONE